# Towards Understanding Hybrid Protein Language Model Design: A Systematic Ablation and Interpretability Study

**Yash Semlani**[*]
Lyra Labs, UNC Chapel Hill
yash@lyralabs.ai

**Nauman Javed**[*]
Lyra Labs
nauman@lyralabs.ai

**Frederick Hoffman**
Lyra Labs, MIT
freddy@lyralabs.ai

**Krithik Ramesh**
Lyra Labs
krithik@lyralabs.ai

## ABSTRACT

Protein language models (PLMs) have emerged as powerful tools for learning sequence representations for diverse downstream prediction tasks and protein design. Recent PLMs which incorporate mixture-of-experts (MoE) layers, state-space models (SSMs), and hybrid SSM-attention architectures have shown strong performance. However, how each of these components impacts performance remains poorly characterized. Here, we systematically evaluate how MoE, SSM, and hybrid architectures—both in isolation and in combination—affect PLM performance. For each component, we examine how key hyperparameters, including hybrid layer ratios and expert sparsity, influence performance. We find that hybrid architectures combining attention and SSM layers consistently outperform parameter-matched non-hybrid baselines. In contrast, incorporating MoE layers improves performance on some tasks while degrading it on others relative to dense baselines. For MoE models, we find that moderate expert sparsity optimizes the tradeoff between effective model capacity and routing stability. Analysis of expert routing in MoE-hybrid models reveals emergent expert specialization aligned with protein secondary-structure characteristics. Together, these findings provide an empirically grounded framework for designing the next generation of high-capacity, compute-efficient PLMs.

## 1 INTRODUCTION

By learning statistical patterns through training on millions of protein sequences, protein language models capture latent representations of structure and function without explicit supervision. Models such as ESM-2 (Lin et al., 2023a), and ProtTrans (Elnaggar et al., 2022) have demonstrated that masked language model pretraining over protein sequences yields embeddings useful for a broad range of downstream tasks.

Simultaneously, the range of architectural choices available for protein language models has grown significantly. Mixture-of-experts layers, which split traditional feed-forward networks into specialized experts of which only a subset are activated per token, have been applied to protein pretraining by models such as AIDO, (Sun et al., 2024), which demonstrated that sparse MoE Transformers can match or exceed dense models of comparable active parameter count. State space models, which offer subquadratic sequence modeling with efficient long-range dependency capture, have been adapted for protein sequences by Lyra (Ramesh et al., 2025), ProtMamba (Sgarbossa et al., 2025), and PTM-Mamba (Peng et al., 2025) — with Lyra in particular arguing that the design of SSMs are well suited to model biological phenomena such as epistasis. Hybrid architectures, which interleave Transformer attention layers with SSM or linear-attention layers within a single model, have emerged as a particularly successful design pattern. By combining the long-range recall of

---

[*]Equal contribution.

attention with the efficient sequence mixing of state-space layers, hybrids can match or exceed pure-Transformer performance at substantially lower computational cost. This paradigm has been validated extensively on language tasks (Lieber et al., 2025; Glorioso et al., 2024; Dao & Gu, 2024), and has recently been adopted for protein modeling by the Dayhoff Atlas (Yang et al., 2025), which uses a hybrid SSM–Transformer–MoE architecture for autoregressive protein generation.

These combinations are motivated by the complementary inductive biases of each primitive: attention explicitly models pairwise interactions between all positions, SSMs capture sequential dependencies with efficient long-range propagation, and MoE introduces per-token conditional computation that may specialize toward distinct structural or functional contexts.

Each line of work has produced compelling results in isolation. However, these studies are typically conducted at different parameter scales, on different pretraining corpora, with different optimization protocols, and evaluated on different downstream benchmarks. As a result, positive results accumulate across the field, but without controlled comparisons, they cannot be decomposed into attributable architectural effects. Moreover, while recent hybrid models have demonstrated that these primitives can be effectively combined, it remains unclear which components drive gains on which tasks, and how mechanisms such as expert specialization in MoE layers shape downstream performance on different tasks.

Here, we conduct a systematic ablation of protein language model architecture, holding pretraining data, training protocol, and evaluation suite constant across all models at a shared 20M active-parameter budget. We compare dense Transformers, Transformers augmented with mixture-of-experts layers at two sparsity configurations, a bidirectional state space model (Hydra), hybrid SSM–Transformer architectures at two block ratios, and hybrid architectures further combined with MoE — evaluating all models across the TAPE benchmarkRao et al. (2019) suite spanning stability, fluorescence, secondary structure, remote homology, and contact prediction. Our contributions are:

1. We develop a controlled experimental framework for comparing hybrid pLM architectures, training all configurations under matched compute and capacity constraints within an ESM-like masked language modeling protocol, varying key hyperparameters for each primitive.

2. We evaluate downstream performance across standard protein property prediction benchmarks, revealing that hybrid SSM–Transformer architectures are the most broadly effective configurations, while MoE introduces a consistent task-dependent trade-off — improving local, per-residue tasks such as stability and secondary structure prediction at the cost of global, relational tasks such as remote homology detection and long-range contact prediction.

3. We investigate expert utilization within MoE layers, showing that experts develop emergent specialization toward local structural features such as secondary structure elements. We also find that auxiliary load-balancing loss during fine-tuning amplifies this specialization with corresponding task-specific performance gains, providing mechanistic insight into the observed pattern of task-dependent improvements and regressions.

## 2 BACKGROUND AND PRELIMINARIES

### 2.1 PROTEIN LANGUAGE MODELS AND MASKED LANGUAGE MODELING

Protein language models treat amino acid sequences as sentences over a 20-letter alphabet and learn representations via self-supervised objectives. ESM-2 (Lin et al., 2023a) uses masked language modeling (MLM): given a sequence $\mathbf{x} = (x_1, \ldots, x_L)$, a random subset of positions is masked, and the model predicts the original amino acid identities from the unmasked context. Training on millions of sequences from UniRef (Suzek et al., 2014) yields representations that encode evolutionary conservation, secondary structure, and inter-residue contact information (Rives et al., 2021). ESM-2 uses a bidirectional Transformer encoder with rotary positional embeddings Su et al. (2023) and has been scaled from 8M to 15B parameters (Lin et al., 2023a).

## 2.2 State Space Models and Bidirectional Mamba

State space models (SSMs) map an input sequence to an output through a latent continuous-time dynamical system, discretized for sequence modeling (Gu et al., 2022). Mamba (Gu & Dao, 2023) introduced input-dependent (selective) parameterization of the SSM matrices, enabling content-aware processing with linear-time complexity. Standard Mamba processes sequences unidirectionally, which is suitable for autoregressive generation but insufficient for MLM where full bidirectional context is needed. Following prior work (Peng et al., 2025; Li et al., 2024), we employ bidirectional Mamba blocks that process sequences in both forward and reverse directions and combine the resulting representations. Mamba-2 (Dao & Gu, 2024) further established a duality between SSMs and structured attention, enabling more hardware-efficient computation.

## 2.3 Mixture of Experts

Mixture-of-experts (MoE) replaces a single feed-forward network with $N$ parallel expert networks and a learned router that selects, for each token, the top-$k$ experts to process that token (Shazeer et al., 2017). This allows scaling model capacity (total parameters) without proportionally increasing compute (active parameters). In hybrid architectures like Jamba (Lieber et al., 2025), MoE layers are interleaved with attention and SSM layers, applied at regular intervals to increase capacity while keeping inference efficient. To encourage even expert utilization during training, Jamba utilizes auxiliary load-balancing losses.

## 3 Methods

All models are trained under a unified ESM-2-style masked language modeling protocol on UniRef50 sequences, with 15% token masking following the BERT corruption scheme (Devlin et al., 2019), for 500K steps at a global batch size of 2,048 sequences (full training details in Appendix A.1). Every variant is constrained to approximately 20M active parameters and a hidden dimension of 512, ensuring that performance differences reflect architectural choices rather than capacity. We construct the following model classes over this shared backbone: (i) a *dense Transformer* baseline matching ESM-2 at 20M parameters; (ii) a *bidirectional SSM* (Hydra; Hwang et al. 2024), which replaces all Transformer layers with quasiseparable SSM blocks that natively support bidirectional context; (iii) *hybrid* models that interleave Hydra and Transformer blocks at two ratios (4H:1T and 2H:2T), applying rotary position embeddings only in the Transformer layers; (iv) *MoE* variants, in which the feed-forward network in each Transformer block is replaced with a SonicMoE layer (Guo et al., 2025) at two sparsity levels (64 experts / top-8 and 8 experts / top-2), with an auxiliary load-balancing loss during pretraining; and (v) *hybrid MoE* models that combine (iii) and (iv), applying MoE only at the Transformer blocks while Hydra blocks retain dense feed-forward layers. The architectural details for each variant are provided in the Appendix A.2.

We evaluate all models on the TAPE benchmark suite (Rao et al., 2019), covering stability, fluorescence, secondary structure prediction (Q3 accuracy on TS115, CB513, CASP12), remote homology detection (family, superfamily, fold), and contact prediction (precision at L/5 for medium- and long-range contacts). Full fine-tuning details, including optimizer settings, task-specific prediction heads, and auxiliary loss handling, are provided in Appendix A.3. For MoE and hybrid MoE variants, we fine-tune both with and without auxiliary load-balancing loss (reduced to weight 0.001) and report the better result per task. Beyond downstream accuracy, we probe the learned representations through an interpretability analysis. We examine expert utilization by computing P(label | expert) for secondary structure labels across MoE layers, sorting experts by their relative helix-versus-strand preference to reveal emergent structural specialization.

## 4 Results

Among the non-MoE baselines, Hydra outperforms the dense Transformer on secondary structure, remote homology, and medium-range contact prediction, while the Transformer holds an edge on stability and long-range contacts. Fluorescence is comparable across both. Neither baseline dominates all tasks, but Hydra's advantages span a wider set of benchmarks.

| | Stab. | Fluor. | Secondary Structure | | | | Remote Homology | | | | Contact P@L/5 | |
|---|---|---|---|---|---|---|---|---|---|---|---|---|
| | | | ts115 | cb513 | casp12 | Overall | Superfam. | Fold | Family | Overall | Med. | Long |
| **Dense** | | | | | | | | | | | | |
| ESM-2 | 0.717 | 0.676 | 0.778 | 0.746 | 0.703 | 0.751 | 0.492 | 0.232 | 0.927 | 0.605 | 0.359 | 0.360 |
| Hydra | 0.703 | 0.670 | 0.789 | 0.753 | 0.713 | 0.760 | 0.492 | 0.243 | 0.936 | 0.611 | 0.410 | 0.345 |
| **Hybrid** | | | | | | | | | | | | |
| 4H:1T Hybrid | 0.689 | 0.677 | 0.786 | 0.761 | 0.698 | 0.763 | 0.484 | **0.249** | 0.933 | 0.608 | 0.397 | 0.377 |
| 2H:2T Hybrid | 0.711 | **0.678** | 0.786 | 0.761 | 0.710 | 0.764 | **0.513** | 0.228 | **0.937** | **0.616** | **0.420** | **0.410** |
| **MoE** | | | | | | | | | | | | |
| 64:8 MoE | 0.734 | 0.675 | 0.783 | 0.756 | **0.721** | 0.761 | 0.445 | 0.228 | 0.926 | 0.586 | 0.377 | 0.321 |
| 8:2 MoE | 0.735 | 0.674 | 0.779 | 0.746 | 0.708 | 0.751 | 0.408 | 0.222 | 0.899 | 0.559 | 0.351 | 0.302 |
| **Hybrid + MoE** | | | | | | | | | | | | |
| 4H:1T Hybrid MoE | **0.748** | 0.678 | 0.786 | 0.763 | 0.709 | 0.766 | 0.482 | 0.243 | 0.926 | 0.603 | 0.419 | 0.339 |
| 2H:2T Hybrid MoE | 0.747 | 0.675 | **0.795** | **0.767** | 0.719 | **0.770** | 0.488 | 0.229 | 0.928 | 0.603 | 0.402 | 0.362 |

Table 1: Full results across all architectural variants. Orange : best among non-MoE models (Dense + Hybrid). **Blue bold** : best overall. Models that are both best overall and among non-MoE models will be colored best overall. MoE variants report the best of with/without auxiliary fine-tuning loss.

Combining SSM and Transformer blocks into hybrid architectures yields the most broadly effective non-MoE models. The 2H:2T hybrid improves over both baselines on secondary structure, remote homology, and both contact prediction ranges, while also achieving the best fluorescence of any non-MoE configuration. The 4H:1T hybrid offers complementary strengths, including the best fold-level remote homology of any model tested. The only task where hybrids fall short is stability, where both trail the dense Transformer.

An interesting pattern emerges when MoE is introduced: in both hybrid and non-hybrid settings, MoE consistently improves stability and secondary structure while degrading remote homology. Applied to the dense Transformer, MoE produces the strongest non-hybrid stability scores and lifts secondary structure on the most challenging split, but remote homology drops substantially. The same trade-off carries over to the hybrid backbone — hybrid MoE models achieve the best stability and secondary structure of any configuration tested, yet remote homology regresses relative to the non-MoE hybrids. The higher-sparsity 64:8 MoE variant is generally the stronger MoE configuration, and thus was chosen to be used in the Hybrid-MoE models.

To understand the effects of MoE models we investigate expert specialization on secondary structure. We plot the probability of predicting a certain label given a specific expert and sort experts within a layer based on the difference between P(Helix — Expert) and P(Strand — Expert). The expert specialization analysis reveals that MoE experts learn to distinguish between secondary structure labels, with clear helix- and strand-preferring experts emerging particularly in deeper layers (Figure 1). Experts are more frequently specialized toward the helix label, consistent with the higher prevalence of alpha helices relative to beta sheets in the pretraining corpus. Specialization increases with layer depth in both the MoE and hybrid MoE models, though the hybrid MoE exhibits this pattern only in its MoE layers.

This per-token specialization aligns with the tasks where MoE helps most: secondary structure prediction (per-residue classification) and stability (influenced by local structural composition). By contrast, MoE degrades remote homology and long-range contact prediction, both of which depend on sequence-spanning relationships rather than local features. This pattern — MoE improving local tasks while degrading global, relational ones — holds consistently across all backbone configurations.

## 5 Discussion

Our results reveal that architectural complementarity—not any single primitive—drives the strongest downstream performance, but that the nature of this complementarity depends on the granularity of the target task.

The consistent advantage of hybrid SSM-attention architectures reflects complementary inductive biases: attention explicitly computes pairwise interactions suited to long-range co-evolutionary couplings, while SSMs propagate information through a compressed latent state that efficiently captures

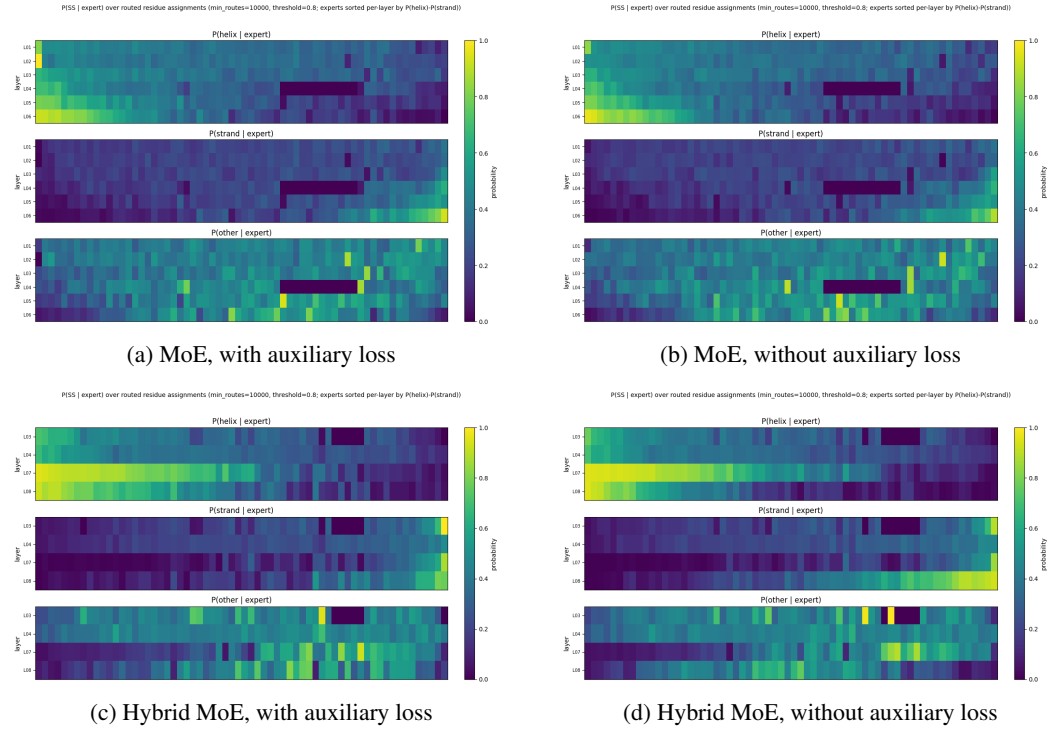

Figure 1: P(Label | Expert) for secondary structure across MoE (top) and Hybrid MoE (bottom) models. Removing the auxiliary load-balancing loss during fine-tuning (right column) sharpens expert specialization, with strand-preferring experts emerging more distinctly.

local sequential dependencies characteristic of secondary structure elements. Proteins are shaped by both pressures simultaneously—local conservation within helices and strands, and global constraints imposed by three-dimensional structure—so architectures combining both primitives outperform those relying on either alone. The balanced 2H:2T hybrid achieves the broadest gains by allocating roughly equal capacity to both regimes, while the 4H:1T hybrid's advantage on fold-level homology suggests that the optimal ratio is task-dependent.

MoE adds a further axis of specialization that is powerful but inherently local. Because routing operates at the token level, experts develop per-residue structural preferences — confirmed by our utilization analysis showing clear helix- and strand-preferring experts that deepen with layer depth. This benefits per-residue and locally determined tasks, with the higher-sparsity 64:8 configuration amplifying gains through finer-grained specialization. However, while standard MLPs are also token-independent, they apply a uniform transformation across positions; MoE replaces this with conditionally selected subnetworks, which may introduce inconsistencies that fragment the coherent sequence-wide representations needed for remote homology detection and long-range contact prediction. That this trade-off holds across all backbone configurations suggests it is likely intrinsic to token-level conditional computation.

The auxiliary load-balancing loss provides further evidence for this interpretation. Removing load balancing during fine-tuning sharpens expert specialization and consistently improves secondary structure prediction, but does not transfer to global tasks—reinforcing that expert specialization is most beneficial when the downstream objective aligns with the granularity of routing decisions. Load balancing thus remains essential during pretraining to prevent expert collapse, but selectively relaxing it during fine-tuning can unlock task-specific gains for per-residue objectives.

## 6 CONCLUSION

We systematically compared SSM, attention, and MoE primitives for protein language modeling under controlled, parameter-matched conditions. Hybrid SSM-attention architectures consistently outperform pure alternatives by combining complementary local and global modeling strengths, while MoE provides targeted gains on per-residue tasks at the cost of global relational ones—a trade-off intrinsic to token-level expert routing. Key directions for future work include scaling these architectures to the 650M–3B parameter range where protein language models are typically deployed, developing sequence- or segment-level routing strategies to address MoE's weakness on global tasks, expanding evaluation to tasks such as fitness landscape prediction and protein–protein interaction modeling, and further exploring expert specialization for different protein characteristics. We hope these findings offer a practical foundation for designing hybrid protein language models that balance multi-scale modeling with compute-efficient conditional computation.

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

## A   APPENDIX

We cover the training and architecture breakdowns in detail in this section.

## A.1 TRAINING

We follow the pretraining protocol of ESM-2Lin et al. (2023b) closely to ensure that observed performance differences reflect architectural choices rather than training recipe. All models are trained with a standard masked language modeling (MLM) objective, where 15% of input amino-acid tokens are selected for corruption: 80% of selected tokens are replaced with a dedicated [MASK] token, 10% are replaced with a random amino-acid token, and the remaining 10% are left unchanged, following the BERT masking policy Devlin et al. (2019). Optimization uses fused AdamW Loshchilov & Hutter (2019) with $\beta_1 = 0.9$, $\beta_2 = 0.98$, $\epsilon = 10^{-8}$, and weight decay 0.01, together with a linear warmup–linear decay schedule: the learning rate warms up over 2,000 steps to a peak of $4 \times 10^{-4}$, then decays linearly to $0.1\times$ the peak over the first 90% of training and is held constant thereafter. Training runs for 500,000 optimizer steps with a global batch size of 2,048 sequences (approximately 2.1M padded tokens per step, approximately 1.05T padded tokens total). Training is distributed across multiple GPUs using PyTorch DDP with bf16 mixed-precision autocast; validation is performed every 10,000 steps. For MoE variants, an auxiliary load-balancing loss is applied during pretraining with weight 0.01. All models are trained on 32 or 64 NVIDIA H100 GPUs (4 or 8 nodes of 8 GPUs each). Input sequences are capped at 1,024 tokens; proteins exceeding this length are cropped by sampling a random start position and extracting a contiguous window of 1,024 tokens, exposing the model to diverse subsequences across training. Following ESM-2, each sequence is prefixed with a BOS/CLS token, and an EOS token is appended only when the full sequence fits within the context window, providing an explicit signal distinguishing complete sequences from cropped fragments.

## A.2 ARCHITECTURES

All models share a common token embedding layer with hidden dimension 512 and are constrained to approximately 20M active parameters. Transformer blocks are implemented using NVIDIA's TransformerEngine library with pre-norm (LayerNorm), rotary position embeddings (RoPE), and no attention or hidden dropout. Hydra blocks use pre-norm with internal RMSNorm and carry no explicit positional encoding beyond the SSM dynamics. The embedding layer applies ESM-style token dropout, zeroing masked-token embeddings and rescaling by the mask ratio to correct the train-time masking mismatch. The MLM prediction head uses a separate decoder projection and is not weight-tied to the input embedding matrix. We train the following variants: Dense Transformer. Our primary baseline is an ESM-2 style encoder only model with 6 layers of multi-headed self-attention followed by a feed-forward network with intermediate dimension 2,048. Sparse Transformer with SonicMoE Guo et al. (2025). We replace the FFN in each Transformer layer with a SonicMoE layer at two sparsity configurations: 64 experts with top-8 routing (64:8) and 8 experts with top-2 routing (8:2). In both cases, total parameters exceed the dense baseline but active parameters per token are matched to 20M. Expert MLPs are bias-free with weights forced to bfloat16. During pretraining, an auxiliary load-balancing loss is summed over MoE layers and added to the MLM loss with weight 0.01. Bidirectional SSM (Hydra) Hwang et al. (2024). We replace all Transformer layers with Hydra blocks, a bidirectional state space model grounded in quasiseparable matrices that extends the selective SSM framework of Mamba to bidirectional contexts without the heuristic wrappers used in prior protein SSM work. Hybrid SSM-Transformer. We interleave Hydra and Transformer blocks in two configurations: an SSM-heavy hybrid (4H:1T ×2) repeating four Hydra blocks followed by one Transformer block twice for 10 total layers, and a balanced hybrid (2H:2T ×2) alternating two Hydra and two Transformer blocks twice for 8 total layers. RoPE is applied only in the Transformer blocks. Both configurations are parameter-matched to the pure baselines. Hybrid SSM-Transformer with SonicMoE. Starting from each hybrid configuration, we replace the FFN in each Transformer block with a SonicMoE layer at the same two sparsity levels (64:8 and 8:2). Hydra blocks retain standard FFNs. This targets increased capacity specifically at the attention layers.

## A.3 FINE-TUNING

All TAPE downstream tasks share a single set of fine-tuning hyperparameters, with the sole exception of batch size for contact prediction. No per-model hyperparameter tuning is performed; every architecture is fine-tuned with the identical configuration described below, ensuring that downstream performance differences reflect pretrained representation quality rather than fine-tuning optimization.

**Optimizer and schedule.** We use AdamW ($\beta_1$=0.9, $\beta_2$=0.999, $\epsilon$=$10^{-8}$) with a peak learning rate of $1\times10^{-4}$. The learning rate follows a linear warmup / linear decay schedule: warmup over 10% of the first epoch's steps to the peak value, then linear decay to $0.1\times$ peak over 90% of total training steps, held constant thereafter. Training runs for a maximum of 100 epochs with early stopping (patience 10 epochs, minimum delta 0.002). All fine-tuning uses bfloat16 mixed precision with no gradient clipping.

**Batch sizes.** A batch size of 32 is used for all tasks except contact prediction, which uses a batch size of 2 due to memory constraints from pairwise feature construction.

**Backbone fine-tuning.** The full backbone is fine-tuned end-to-end; no parameters are frozen. All model parameters (backbone and task head) are passed to a single optimizer with a uniform learning rate. No differential learning rates or separate parameter groups are used.

**Task-specific prediction heads.** Each TAPE task uses a lightweight head attached to the pre-trained backbone:

- **Fluorescence and stability** (sequence-level regression): a single linear layer over the CLS token representation ($d_{\text{hidden}} \to 1$). Loss: MSE.
- **Remote homology** (sequence-level classification, 1,195 classes): a two-layer MLP over the CLS token — $d_{\text{hidden}} \to 512$ with GELU activation, then $512 \to 1{,}195$. Loss: cross-entropy.
- **Secondary structure** (per-token classification, 3 classes): a single linear layer over per-token representations ($d_{\text{hidden}} \to 3$). Loss: cross-entropy with padding tokens ignored.
- **Contact prediction** (pairwise classification): pairwise features are constructed from per-token representations via elementwise product and difference, concatenated to form a $2d_{\text{hidden}}$-dimensional vector per residue pair, then projected to 2-class logits via a single linear layer. The output is symmetrized: $(\mathbf{L} + \mathbf{L}^\top)/2$.

All heads use dropout probability $p$=0.0. Head weights are initialized from a normal distribution with standard deviation equal to the model's initializer range.

**Auxiliary loss during fine-tuning.** For MoE and hybrid MoE architectures, the load-balancing auxiliary loss is retained during fine-tuning but reduced by $10\times$ relative to pretraining (weight 0.001 vs. 0.01). For non-MoE architectures, the training objective is simply the task loss. As described in Section 4, we additionally evaluate MoE variants with the auxiliary loss fully removed during fine-tuning and report the better result per task.

