# OpenReview forum: "Towards Understanding Hybrid Protein Language Model Design: A Systematic Ablation and Interpretability Study"
_ICLR.cc/2026/Workshop/FM4Science — ICLR 2026 Workshop FM4Science Poster_

### Official Review · Reviewer_qMco · 2026-02-23
**This paper presents a controlled ablation study of three increasingly popular architectural “primitives” in protein language modeling—state-space models (SSMs), attention–SSM hybrids, and mixture-of-experts (MoE)—under a matched pretraining protocol and a fixed ~20M active-parameter budget. Using ESM-2-style MLM pretraining on UniRef50 and evaluation on the TAPE suite, the authors report that hybrid SSM–Transformer models provide the most consistently strong performance across tasks, while MoE introduces a clear task-dependent trade-off: improvements on local/per-residue tasks (stability, secondary structure) but degradation on global/relational tasks (remote homology, long-range contact prediction).**

**Rating:** 6
**Confidence:** 4

**Review:**

## Strengths

1. Controlled experimental design (major strength)
  The paper’s primary value is methodological rigor in comparison: matched pretraining data (UniRef50), protocol, compute/capacity constraints (~20M active params), and a consistent downstream suite (TAPE). This addresses a real reproducibility gap in PLM architecture claims .

2. Clear, consistent empirical pattern with actionable takeaways
  The conclusion that *hybrids are broadly best* while *MoE induces a local-vs-global task trade-off* is both consistent in the presented table and practically meaningful for designers of compute-efficient PLMs .

3. Interpretability analysis adds mechanistic insight
  The expert specialization analysis (helix/strand preference emergence, increasing with depth) is a useful attempt to connect MoE routing behavior to downstream task outcomes, beyond reporting raw metrics .

## Weaknesses

1. Novelty is primarily empirical/diagnostic rather than methodological
  The paper does not introduce new architectures, routing mechanisms, or training objectives. The contribution is a careful ablation and analysis at a single parameter scale. This is valuable, but for a conference track emphasizing algorithmic novelty, it may be viewed as incremental.

2. Baseline set is incomplete relative to today’s PLM landscape
  Within the chosen primitives (Transformer / Hydra / hybrid / MoE), the study is thorough. However, the comparison does not include other strong PLM baselines or design choices that could materially affect conclusions, e.g., alternative hybridization strategies, different SSM families, or other efficiency mechanisms. This limits how far one can generalize “hybrid is best” beyond the specific Hydra+Transformer instantiation at 20M.

3. Single-scale evaluation weakens claims about “next generation” design
  All results are at ~20M active parameters. The authors acknowledge that future work should scale to 650M–3B regimes . Given that many architectural effects (especially MoE routing stability, emergent specialization, and long-range behavior) can change with scale, the current conclusions may not transfer cleanly.

---

### Official Review · Reviewer_KKvA · 2026-02-24
**Well designed ablation study on protein language model architectures but limited by small scale**

**Rating:** 6
**Confidence:** 3

**Review:**

This paper systematically compares architectural primitives for protein language models including dense Transformers, bidirectional SSMs via Hydra, hybrid SSM Transformer models at two ratios, MoE at two sparsity levels, and hybrid plus MoE combinations. All variants are trained at 20M active parameters under identical conditions using ESM 2 style MLM on UniRef50 and evaluated on TAPE. The key findings are that hybrid architectures consistently outperform non hybrid baselines, MoE helps local per residue tasks like secondary structure and stability but hurts global tasks like remote homology and contact prediction, and expert routing analysis shows emergent specialization aligned with secondary structure that sharpens when auxiliary load balancing is removed during finetuning. The controlled experimental design is the papers main strength as it isolates architectural effects by matching compute, data and training protocol across all variants. The expert utilization analysis in Figure 1 is also a nice touch. However the 20M parameter scale is a major limitation since protein language models are deployed at 650M to 3B parameters and tradeoffs at this scale may not transfer. The MoE degradation on global tasks could easily be an artifact of tiny expert networks at small scale. The absolute performance gaps in Table 1 are often quite small and no variance or significance tests are reported which makes it hard to confidently attribute differences to architecture rather than noise. Only two hybrid ratios are explored leaving most of the design space unexamined. The mechanistic explanation for why MoE hurts global tasks is reasonable but speculative and not directly verified beyond task level correlations. No comparison to published baselines like ESM 2 8M or 35M is provided which makes it hard to contextualize the numbers.

---

### Official Review · Reviewer_W8P2 · 2026-02-24
**Controlled Architectural Comparison of Protein Language Models with Opportunities for Stronger Statistical and Benchmark Validation**

**Rating:** 6
**Confidence:** 5

**Review:**

The manuscript addresses an important issue in modern machine learning: the lack of unified benchmarks for systematically comparing different model architectures. The authors construct and pre-train several deep learning variants (Transformer-based, SSM-based, hybrid, and MoE models) under a controlled setup, and subsequently evaluate them using lightweight fine-tuning heads on downstream tasks. The paper is well structured and generally easy to follow. However, additional analyses would strengthen the architectural claims and improve the robustness of the conclusions.

Comments
1.	Compared to models such as ESM-2, which were trained for substantially longer, it would be helpful to clarify how the choice of 500K training steps was determined. In particular, was convergence assessed for all architectures? If some models converged substantially earlier, it is possible that continued training may have led to overfitting or performance degradation, potentially affecting the fairness of the comparison. Providing training curves or convergence diagnostics would strengthen confidence in the reported results.

2. It appears that downstream results are based on single fine-tuning runs. This limits the statistical reliability of comparisons between models, especially when performance differences are modest. Reporting mean and standard deviation across multiple fine-tuning seeds would allow for a more rigorous comparison and enable statistical testing of architectural differences.

3.	Given the known sensitivity of Mixture-of-Experts (MoE) models to initialization and routing dynamics (e.g., arXiv:2202.08906), it would strengthen the architectural claims to report variability across backbone pretraining runs. MoE training can exhibit seed-dependent behaviors, including differences in expert utilization, load balancing stability, and emergent specialization patterns. Since the paper analyzes expert specialization and performance trade-offs, demonstrating that these effects are consistent across multiple pretraining replicas, or at least discussing potential variability, would increase confidence that the observed behaviors reflect architectural properties rather than single-run stochastic effects.

4.	The evaluation is conducted exclusively on TAPE. While TAPE is widely used and appropriate for controlled comparison, it primarily emphasizes structural and biophysical tasks (e.g., secondary structure, contact prediction, remote homology). Given that the paper makes broader architectural claims about protein foundation models, it would strengthen the generality of the conclusions to either include additional functional or interaction-focused benchmarks, or to more explicitly discuss how the observed trade-offs (e.g., MoE gains versus remote homology regression) might extend beyond the TAPE setting.

Minor Comment: Figure 1 axis labels are very difficult to read

---

### Official Review · Reviewer_qxJr · 2026-02-24
**well-controlled architectural study**

**Rating:** 7
**Confidence:** 2

**Review:**

This paper compares different architectures for protein language models under carefully controlled conditions. All models are trained using the same data, training setup, and parameters, allowing the authors to study how architecture alone affects performance.
The study evaluates several model types: standard transformers, state space models, hybrid models, and MoE variants. Models are tested on the TAPE benchmark, which includes tasks such as protein stability prediction, secondary structure prediction, fluorescence prediction, remote homology detection, and contact prediction.

The results show that hybrid models combining state space layers and attention layers perform best as they capture both local sequence patterns and long-range relationships. Adding MoE layers improves tasks that depend on local features but harms tasks requiring global understanding.

Pros:
- Nice comparison: all models use the same training setup, making results reliable.
- Clear demonstration that hybrid architectures work better than other architectures used for comparison.
- Provides useful insight into when MoE helps and when it hurts performance.

Cons:
- Experiments are limited to relatively small models.
- Only a few hybrid configurations are tested.
- Biological interpretation, practical implications could be expanded further.

---

### Meta-Review · Area_Chair_jPsr · 2026-02-28

**Recommendation:** Accept (Poster)
**Confidence:** 4

**Metareview:**

The reviewers all agreed that this work presents a well designed and controlled empirical evaluation of architecture choices for protein language models (PLMs). Expanding the scope and scale of the evaluation will strengthen the work in the future.

---

### Decision · Program_Chairs · 2026-03-03

Accept (Poster)